# OpenReview forum: "Zeroth-Order Forward-Only SNN Training Inspiring Neuromorphic On-Chip Learning"
_ICML.cc/2026/Conference — ICML 2026 regular_

### Official Review · Reviewer_j9yC · 2026-03-09

**Soundness:** 3
**Presentation:** 3
**Significance:** 3
**Originality:** 3
**Overall Recommendation:** 4
**Confidence:** 3

**Summary:**

To reduce the high costs of Backpropagation (BP) in on-chip learning, this paper presents a Zeroth-Order (ZO) optimization framework for Spiking Neural Networks (SNNs). Theoretical and empirical analyses show that the discrete Heaviside step function in SNNs leads to much higher variance in ZO gradient estimation than smooth activations like ReLU. To solve this, the study introduces Subspace Zeroth-Order (SZO) optimization. This method projects random perturbations into a low-dimensional subspace of historical gradient trajectories to suppress noise. SZO achieves competitive, forward-only performance on ImageNet-1K and shows strong robustness in continual learning.

**Compliance With Llm Reviewing Policy:**

Affirmed.

**Final Justification:**

Most concerns have been resolved, and I have decided to maintain the weak acceptance score.

**Key Questions For Authors:**

1. Can the SZO method build a good subspace from scratch without using any BP-related data at the beginning?

2. How does the subspace method handle the multiple time-steps in SNNs without losing gradient accuracy?

3. What are the actual energy savings and speedups when this algorithm is deployed on real neuromorphic chips like Loihi or Tianjic?

**Limitations:**

No. This work could consider appending a limitations section.

**Strengths And Weaknesses:**

**Strengths**:

1. It explains why standard Zeroth-Order (ZO) methods fail in SNNs due to high variance from the Heaviside step function.

2. It is the first forward-only SNN training method to perform well on large-scale datasets like ImageNet-1K.

3. The method uses forward-only passes, which better match how the human brain learns compared to BP.

4. It significantly reduces on-chip memory usage because it does not need to store activation values for a backward pass.


---


**Weaknesses**:

1. The method needs historical gradient trajectories to build the subspace. This might still require some initial BP training.

2. Storing the projection matrices for the subspace adds new memory costs to the hardware.

3. The performance depends heavily on the perturbation scale and the dimension of the subspace.

4. The paper should compare more with other bio-plausible methods like Direct Feedback Alignment (DFA) or e-prop.

---

> ### Author Rebuttal · Authors · 2026-03-31
>
> We sincerely appreciate the reviewer for the valuable comments and constructive suggestions.
> ## 1. The method needs historical gradient trajectories to build subspace...
> Thank you for the comment. Our target scenario is on-chip learning from a pretrained model, and the subspace can be constructed offline during pretraining. The deployed model is then updated by SZO using forward-only learning. A typical application is unsupervised online adaptation in cross-domain scenarios, see Sec. 6.3.
>
> This setting echoes the brain’s learning paradigm, where innate initialization is followed by efficient adaptation, and we hope it can inspire scalable neuromorphic on-chip learning in real-world settings.
> ## 2. Storing the projection matrices for subspace adds memory costs.
> Our analysis in Figure 5 shows that **while the memory of BP and BP-mix grows steeply with both batch-size and timestep, that of SZO increases only marginally w.r.t batch-size and time-step**.  Though SZO needs to store projection matrix, this overhead is much smaller than the activation storage required by BP and BP-mix, thus has **substantially lower overall memory overhead**.
> ## 3. Performance depends on perturbation scale and subspace dimension.
> Our ablation in Sec. 6.5 shows that SZO performs well across a fairly broad range of perturbation scales (Table 4), and increasing the subspace dimension generally improves accuracy (Figure 6). When $k=60$, the accuracy is close to that of BP. A subspace with only **a few tens of dimensions** is sufficient for most cases.
> ## 4. Compare with bio-plausible methods like DFA or e-prop.
> Thanks for this constructive suggestion. We conducted experiments to compare with DFA and OPZO [1]. OPZO is a pseudo-ZO algorithm that also uses direct error feedback to each layer.
>
> We consider both the Spike-CNN model in [1] and other 3 models in our work. As shown in Table 1, OPZO performs well on shallow CNN and VGG, but is limited on deep ResNets where SZO significantly outperforms OPZO and DFA.
>
> Table 2 compares SZO with OPZO in finetuning ResNet-34 on ImageNet in the presence of Gaussian noise. Clearly, SZO outperforms OPZO. We also compared with EqSpike [2], see the second reply to Review-1(Kj3r). Code for reproducing these results is available at the link in our paper.
>
> Table 1. Acc (%) on the training from scratch task.
> |Dataset| Model |Params|BP|BP-Subspace|SZO-CGE|SZO-RGE(q=1)|SZO-RGE(q=20)|OPZO|DFA|
> |-|-|-|-|-|-|-|-|-|-|
> |CIFAR10| Spike-CNN (5-layer)|3.92M|90.79|90.78|90.78|90.67|90.72|89.42|82.38|
> |CIFAR100| Spike-CNN (5-layer)|4.66M |67.17|67.40|67.42|67.32|67.03|64.77|54.76|
> |CIFAR10| Spike-ResNet20 |0.27M |86.74|86.42|86.21|81.29|83.71|54.45|38.79|
> |CIFAR100| Spike-ResNet18|11.22M|74.51|73.32|73.35|73.77|73.79|36.90|28.36|
> |DVS-Gesture| Spike-VGG11|9.50M|97.57|96.65| 96.65|91.78|95.72|96.06|91.43|
>
> Table 2. Acc (%) for fine-tuning ResNet-34 on ImageNet under different noise scale.
> |Noise scale|No finetuning|SZO-CGE|SZO-RGE(q=10)|OPZO|
> |-|-|-|-|-|
> |0.1|60.29|64.24|64.54|63.39|
> |0.15|56.21|64.07|64.08|60.96|
> ## 5. Can build a subspace from scratch without using BP...?
> We have tried random subspace but have not achieved satisfactory performance yet. Building an effective subspace from scratch remains an important direction deserves future study.
> ## 6. How does the method handle the multiple time-steps...?
> SZO applies perturbation to the weights. In SNNs, the model weights are shared across all timesteps. Therefore, perturbing the weights affects the network dynamics of all timesteps and finally the model output. Our evaluations on static datasets with 4 timesteps and on DVS dataset with 20 and 80 timesteps demonstrate the effectiveness of this approach.
> ## 7. Actual energy savings and speedups when deployed on real neuromorphic chips?
> Thanks for this important question. We have not yet conducted deployment on neuromorphic chips. Our contribution is primarily **algorithmic**: SZO removes the main bottlenecks that make BP unsuitable for on-chip learning. SZO can reduce energy and improve speed on neuromorphic hardware as (detailed in Section 7 and Appendix B):
> 1. It avoids backward computation and activation buffering, which reduces memory traffic and control complexity. Consequently, SZO has much lower runtime memory than BP (Figure 5).
> 2. SZO performs perturbation and optimization in a **low-dimensional subspace** with $k \ll d$, which greatly reduces the cost of generating and manipulating perturbations.
> 3. SZO converges substantially faster than full-space BP (Figure 3 and 4),  which is desirable for real-time on-chip learning, e.g., on ImageNet-1K SZO reaches similar accuracy in <10 epochs v.s. about 60 epochs for BP.
>
> We view deployment on neuromorphic chips as an important future direction.
> ## 8. consider a limitation section.
> We will add a limitation section in the revision.
> ## References
> [1] Online pseudo-zeroth-order training of neuromorphic SNNs. ICLR, 2026
>
> [2] EqSpike. iScience, 2021

---

> > ### Author Rebuttal · Reviewer_j9yC · 2026-04-03
> >
> > Thanks for your detailed response. Most concerns have been resolved, and I will maintain the weak acceptance score.

---

> > > ### Author Response · Authors · 2026-04-08
> > >
> > > We thank the reviewer for the insightful comments, which help us improve the paper: to clarify the reliance on a BP stage for obtaining the subspace, to compare with bio-plausible approaches such as DFA, and to clarify how the method handles multiple time steps, as well as its potential energy savings and speedups when deployed on real neuromorphic chips. We are glad that our responses have addressed most of your concerns. Thanks for your time and effort in reviewing our paper.

---

### Official Review · Reviewer_abx4 · 2026-03-10

**Soundness:** 2
**Presentation:** 3
**Significance:** 2
**Originality:** 2
**Overall Recommendation:** 2
**Confidence:** 4

**Summary:**

The paper mainly proposes a parameter optimization method using subspaces in SNN without BP participation. Firstly, the paper analyzes the weaknesses of the original CGE method in SNN, mainly the high initial variance and the problem that simple perturbations cannot affect the model output. Then, it points out the low dimensional characteristics of the neural network training trajectory, and considers using CGE/RGE method in low dimensions to achieve computation. The paper demonstrates that when the low dimensional k<<d, the model convergence speed is significantly improved. Subsequently, validate the argument on relevant tasks

**Compliance With Llm Reviewing Policy:**

Affirmed.

**Final Justification:**

The author has resolved most of my questions; I will lower my rating.

**Key Questions For Authors:**

1. The background of the paper mentions the performance advantages of CGE compared to RGE, while Table 3 in the main text shows that the most advantageous method is actually RGE's q=1 method, which theoretically produces greater variance changes.How to explain?
2. The original theoretical assumption is to perturb the entire space, and the paper uses PCA to perform subspace dimensionality reduction, which directly determines the performance representation based on the subspace dimension k, making it difficult to guarantee k<<d, there is no corresponding explanation on how to select the k value in the experiment?
3. The extensive forward calculation, PCA process, and reverse BP process required to preserve the parameter transformation matrix in the core of the paper cannot be applied to the current hardware foundation, making it impossible to achieve the maximum goal of the article

**Limitations:**

The paper may need to consider other usage scenarios, such as fine-tuning on the end side, Considering the large number of BP dependencies and complex PCA operations involved in the entire usage process, the current on-chip operations are unable to hold true

**Strengths And Weaknesses:**

strengths:
1. The viewpoint proposed in the paper that SNN learns on chips is an important issue in neural morphology
2. The analysis and observation of the paper are sufficiently clear, clearly explaining the motivation behind the work of the paper

weaknesses:
1. The method actually relies on BP training trajectories to construct subspaces, which directly affects the original conclusion that does not require BP
2. The experimental comparison in the paper is unfair, as it does not take into account the computational complexity required to obtain subspaces
3. The method proposed in the paper is still implemented on GPU without any hardware.

---

> ### Author Rebuttal · Authors · 2026-03-31
>
> We sincerely thank the reviewer for the valuable comments.
> ## 1. (1) Relies on BP to construct subspaces. (2) does not account computational complexity required for subspaces. (3) extensive forward calculation, PCA, and reverse BP... cannot be applied to hardware... (4) consider scenarios, such as fine-tuning on the end side...
> Thanks for the comment. We apologize for not clarifying the distinction between the **offline subspace construction and the online on-chip learning stages** in our paper.
>
> Our target scenario is **on-chip learning based on a pretrained model**. This setup echoes the brain's learning paradigm, where neural architectures and synapse strengths are innately initialized, and only modest plasticity is required to acquire new tasks. In our settings, subspace is built offline, and **in on-device learning stage, learning is performed upon pretrained models by forward-only gradient estimation in the fixed subspace**.
>
> We agree that considering **end-side fine-tuning** is highly relevant. In fact, this is the application in our unsupervised model adaptation experiments (Sec. 6.3), where a pretrained model is unsupervisedly adapted to test data distribution shifts (e.g., corruptions). **The subspace is constructed offline with one representative corruption (e.g., Gaussian noise), and then used by SZO to adapt the model to unseen corruptions on-device**.
>
> **While ZO methods alone cannot effectively train large networks (especially SNNs) from scratch, our results demonstrate that SZO using a pretrained model and subspace is highly effective for adaptation** (much **faster** than full-space BP, see Figure 3-4), This validates its potential for on-chip learning with pretraining in the cloud and adaptation on the edge.
> ## 3. The method is still implemented on GPU without any hardware.
> Our work focuses on addressing the **algorithmic** bottlenecks that make BP unsuitable for on-chip learning. For the first time, we show ZO can train deep SNNs at scale, and SZO is **forward-only** that offers strong suitability for practical on-chip learning:
> 1. No dedicated feedback circuits.
> 2. No need to store activations, thus has significant lower memory overhead than BP.
> 3. Unlike vanilla ZO methods that require repeatedly sampling high-dimensional random vectors of size $d$, SZO only samples within a subspace of dimension $k$ (typically tens), **which not only accelerates optimization but also removes the hardware bottleneck of generating high-dimensional random perturbations**.
> 4. Learning in a subspace enables significantly **faster convergence** compared to full-space BP (Figure 3-4), which is desirable for real-time on-chip learning.
>
> We provided detailed analyses that directly reflect on-chip constraints in Sec. 6.4, such as (i) GPU runtime memory under varying batch size and timesteps, which shows much lower memory usage than BP (Figure 5), (ii) comparison of wall-clock time and convergence speed, where SZO converges in fewer epochs (Table 3). More detailed discussion see Sec. 7 and Appendix B.
> ## 4. Table 3 the most advantageous method is RGE's q=1, which theoretically produces greater variance. How to explain?
> Thanks for your question. CGE typically outperforms RGE in high-dimensional settings. However, SZO operates in a very low-dimensional subspace (tens), which largely eliminates the difference between SZO-CGE and SZO-RGE.  As shown in Table I and II, for full-space ZO, the gradient variance is on the order of $10^{4} \sim 10^{5}$, whereas after restricted in subspace, the variance of SZO-CGE, SZO-RGE drop to a much smaller scale ($10^{-3} \sim 10^{-1}$). Therefore, although SZO-RGE ($q = 1$) is noisier, in the low-dimensional subspace this variance gap no longer dominates performance and has only a minimal practical effect.
>
> Table I. Gradient variance of full-space ZO (RGE, $q$=20) on CIFAR10 with SpikeResNet20.
> |Epoch|20|40|60|80|100|
> |-|-|-|-|-|-|
> |ZO|45447.1|43637.6|53787.6|52355.2|62375.2|
>
> Table II. Gradient variance of SZO on CIFAR10 with SpikeResNet20.
> |Epoch|10|20|30|40|
> |-|-|-|-|-|
> |SZO-CGE| 0.0074|0.0073 |0.0066|0.0062|
> |SZO-RGE($q$=1)|0.0787|0.2527|0.1573|0.1041|
> |SZO-RGE($q$=20)|0.0166|0.0153|0.0157|0.0153|
> ## 5.  How to guarantee k<<d, how to select k for PCA-based subspace.
> Since the subspace is constructed via PCA from a sampling window of size $N$, we have $k \leq N$. In our experiments, $N \leq 150$, while $d$ is the full parameter dimension, so $N \ll d$, and $k \ll d$.
>
> Figure 6 shows that performance generally improves as $k$ increases. But when $k = 60$, the accuracy is already close to BP, and further gains are marginal. Additionally, we evaluated SZO with different $k$ on the ImageNet-1K training from scratch task (Table III) and observed a similar trend. Therefore, choosing $k$ in the order of a few tens is sufficient for most cases.
>
> Table III. Acc of SZO-RGE($q$=1) with different subspace dimension $k$.
> |$k$|20|40|60|80|
> |-|-|-|-|-|
> |Acc(%)|57.78|61.33|61.78|62.00|

---

> > ### Author Rebuttal · Reviewer_abx4 · 2026-04-02
> >
> > The author needs to add the acquisition time of subspaces, especially when the model becomes larger and the overall PCA time is unclear.
> >  When the usage scenarios are restricted, the content of the article does indeed become a possibility for use.

---

> > > ### Author Response · Authors · 2026-04-02
> > >
> > > We sincerely thank the reviewer for the further comment. Here, we provide explanation and results to respond to your further concern about **"the acquisition time of subspaces, especially when the model becomes larger and the overall PCA time is unclear."**
> > > ### (1) The PCA computation time
> > > Generally, in our implementation, the subspace extraction by PCA does not consume too much time, even when the model becomes larger. Specifically, given $N$ model weights $\{ {{{ \theta}_1},{ \theta}_2...,{ \theta}_N } \}$, we perform centered PCA. Denote
> > >
> > > $\Theta=[ {{{\theta}_1 - {\bar \theta}},{\theta}_1- {\bar \theta},...,{\theta}_N - {\bar \theta}} ]\in \mathbb{R}^{{d} \times N}$,
> > >
> > > where $\bar \theta  = \frac{1}{N}\sum\nolimits_{i = 1}^N {{\theta _i}}$ and $d$ is the model parameter dimension. Our goal is to compute the $k$ eigenvectors $P=[{v}_1, {v}_2, ..., {v}_k]\in \mathbb{R}^{{d} \times k}$ corresponding to the top-$k$ largest eigenvalues of $\Theta\Theta^\top\in \mathbb{R}^{{d} \times d}$. However, a naive spectral decomposition on $\Theta\Theta^\top\in \mathbb{R}^{{d} \times d}$ to extract $P$ is prohibitively expensive for large $d$.
> > >
> > > Therefore, in implementation, we extract the projection matrix $P$ from $\Theta\in \mathbb{R}^{{d} \times N}$ by a fast computation approach. Specifically, since $N \ll d$, we first perform spectral decomposition on the much smaller matrix $\Theta^\top\Theta\in \mathbb{R}^{{N} \times N}$ to obtain its eigenvectors $[u_1,u_2,\cdots,u_N]\in \mathbb{R}^{{N} \times N}$ and the corresponding singular values $[\sigma_1,\sigma_2,\cdots,\sigma_N]$. Then, the principal directions in the parameter space are recovered as $v_i=\Theta u_i / \sigma_i$, respectively. This approach largely reduces the computational cost.
> > >
> > > We evaluated the PCA computation time for subspace construction under the training-from-scratch task. The experiments were conducted on Intel Xeon Platinum 8280 CPU. As shown in Table 1 below, the PCA computation takes less than 5 seconds across different models, which indicates that the overhead of subspace construction is very small in practice. In the unsupervised model adaptation tasks, both $N$ and $k$ are smaller ($N=40$, $k=20$), so the PCA computation time for subspace construction is even lower. We will add the PCA computation time in the revised manuscript.
> > >
> > > Table 1. PCA computation time for subspace construction
> > > | Model|Param |Sampled Models ($N$) |Subspace Dimension ($k$) | Time (seconds)|
> > > |-|-|-|-|-|
> > > |Spike-VGG11| 9.50M|80 | 20 | 0.76s |
> > > |Spike-ResNet18|11.22M | 150 | 60 |2.32s|
> > > |Spike-ResNet50|25.56M | 150 | 60 | 4.91s |
> > > ### (2) The sampling of the models used for PCA subspace construction
> > > 1. In the training-from-scratch task, we use $N=150$ models for ImageNet-1K, which are sampled over 60 epochs of the pre-training process. For CIFAR-10/100, we use $N=150$ models, which are sampled over 150 epochs of the pre-training process. For DVS-Gesture, we use $N=80$ models, which are sampled over 80 epochs of the pre-training process.
> > > 2. In the **unsupervised model adaptation task, which is highly relevant to our considered on-chip learning scenario**, we use $N=40$ models. Specifically,
> > > * For ImageNetC, we sample 20 BP-trained models in 20 supervised training epochs on Gaussian corruption, with 50000 samples per epoch . Then, these models are combined with another 20 source models (sampled within 8 epochs in the source domain training) to extract a $k=20$ dimensional subspace using PCA.
> > > * For CIFAR-10C/100C, we sample 20 BP-trained models in 10 supervised training epochs on Gaussian corruption, each epoch with 10000 samples. Then, these models are combined with another 20 source models (sampled within 20 epochs in the source domain training) to extract a $k=20$ dimensional subspace using PCA.
> > >
> > > We will make these points clearer in the revised manuscript.
> > >
> > > Once again, we would like to thank the reviewer for the valuable and insightful comments. In particular, the comments help us to clarify the usage scenarios of our method and the scope of our work, the role of the BP pretraining stage in obtaining the subspace, and why RGE outperforms CGE in the experiments. These comments are constructive in improving our paper.  Thanks for your time and effort in reviewing our paper.

---

### Official Review · Reviewer_x3o4 · 2026-03-12

**Soundness:** 3
**Presentation:** 3
**Significance:** 3
**Originality:** 3
**Overall Recommendation:** 5
**Confidence:** 3

**Summary:**

This study investigates if zeroth-order (ZO) optimization, which does not require backpropagation, can be used to train spiking neural networks (SNNs). The authors found that SNNs are more sensitive to inputs, making the variance of ZO gradient estimates much higher in SNNs than in ANNs and thus making ZO less effective in SNNs. To address this issue, they proposed to use ZO optimization to estimate the weight updates in the subspace identified by PCA. Once the weight updates are evaluated in the subspace, they are converted back to the full weight space where the full weights are updated.

**Compliance With Llm Reviewing Policy:**

Affirmed.

**Key Questions For Authors:**

How many time steps are used to accumulate the spikes during inference?

**Limitations:**

Yes

**Strengths And Weaknesses:**

Strengths

The presented idea is novel, and their empirical evaluation supports that their algorithm (SZO) is comparable to backpropagation in terms of model accuracy of spiking neural networks. Since SZO also reduces the complexity of hardware required for training, it may provide an effective solution for on-chip learning.

Weaknesses

1. The accuracy of the model trained by SZO is substantially lower than that of the model constructed by ANN-to-SNN conversion. Furthermore, ANN-to-SNN conversion algorithm can reduce the complexity of neuromorphic chips, as it does not need to estimate gradients. Then, my question is, what kind of advantage does the on-chip learning algorithm have compared to ANN-to-SNN conversion algorithms?

2, The authors do not clearly show how many training runs were used to evaluate SZO. In most of the plots, the error bars are missing. There are error bars in Table 1 and 4, but they do not explain how they are estimated.

---

> ### Author Rebuttal · Authors · 2026-03-31
>
> We sincerely appreciate the reviewer for the valuable feedback. Below we provide explanations to address your comments.
> ## 1. ... What kind of advantage does the on-chip learning algorithm have compared to ANN-to-SNN conversion algorithms?
> We thank the reviewer for this insightful question. We agree that ANN-to-SNN conversion can achieve higher accuracy. However, ANN-to-SNN conversion mainly targets **offline SNN construction**: it first requires a high-performance ANN and then converts it into an SNN, and typically requires a large number of timesteps to approximate the original ANN [1]. In contrast, our work focuses on **on-chip online learning after deployment**, where the key requirement is efficient model updating on neuromorphic hardware using **forward-only** computation, without backpropagation or feedback circuits. This is exactly the setting emphasized in our paper.
>
> Therefore, the main advantage of on-chip learning algorithm is **post-deployment adaptability**. As shown in our experiments, SZO supports continual training and unsupervised online adaptation from a pretrained model. For example, in the unsupervised online adaptation experiments under domain shifts (noise corruptions), **a pretrained model can be continuously adapted online to changing corruption scenarios with the efficient forward-only SZO algorithm**, which can substantially improve the cross-domain performance. Under the continual adaptation experiments in the presence of severe noise corruption (Table 2), SZO-CGE improves the acc. from 45.01% to 76.10% on CIFAR10C, from 29.81% to 79.69% on CIFAR100C, and from 9.36% to 33.37% on ImageNetC.
>
> ANN-to-SNN conversion, by contrast, does not naturally support this kind of continuous on-chip learning, since it relies on first training an ANN offline and then converting it.
> ## 2. The authors do not clearly show how many training runs were used to evaluate SZO. In most of the plots, the error bars are missing. There are error bars in Table 1 and 4, but they do not explain how they are estimated.
> We are sorry for this confusion. In Table 1 and 4, we run each method 5 times independently with different random seeds and report the results as mean ± standard deviation. We will clarify this in the revised version. Training curves with error bars are provided in the anonymous link [2] (see Figure 1 and 2 therein), and we will include them in the revised version.
>
> ## 3. How many time steps are used to accumulate the spikes during inference?
> Thanks for the comment. During inference, we accumulate the output spikes over all $T$ time-steps and use the averaged response as the final prediction:
> $$
> \hat{\mathbf{y}}=\frac{1}{T}\sum_{t=1}^{T}\mathbf{o}^{(t)},
> \qquad
> \mathrm{Pred}=\arg\max_{c}{\hat{y_c}}.
> $$
> Here, $\mathbf{o}^{(t)} \in \mathbb{R}^{C}$ denotes the output vector at time-step $t$, $C$ is the number of classes, $\hat{\mathbf{y}} \in \mathbb{R}^{C}$ is the averaged output over all time-steps, and $\hat{y}_{c}$ is the $c$-th element of $\hat{\mathbf{y}}$. The final predicted label is determined by the index of the maximum response. We will add explanation on this in the revised version.
>
> ### References
> [1] Optimal ANN-SNN Conversion for High-accuracy and Ultra-low-latency Spiking Neural Networks. ICLR 2022.
>
> [2] [https://anonymous.4open.science/r/SZO-RE-C7EB/](https://anonymous.4open.science/r/SZO-RE-C7EB/)

---

> > ### Author Rebuttal · Reviewer_x3o4 · 2026-04-03
> >
> > Thank you for the clarification. My concerns have been properly addressed.

---

> > > ### Author Response · Authors · 2026-04-08
> > >
> > > We sincerely thank the reviewer for the valuable comments as well as the time and effort devoted to evaluating our work. We especially appreciate the important questions on the distinction from ANN-to-SNN conversion, the estimation of error bars, and the inference time-step accumulation, which help us improve the clarity of both the motivation and the experimental presentation of the paper. We are also grateful for the reviewer's follow-up acknowledgement that the concerns have been properly addressed.

---

### Official Review · Reviewer_Kj3r · 2026-03-13

**Soundness:** 3
**Presentation:** 3
**Significance:** 4
**Originality:** 4
**Overall Recommendation:** 4
**Confidence:** 4

**Summary:**

The paper proposes subspace based zeroth order optimization method for forward-only training on spiking neural networks. The paper investigates the concept of variance amplification induced by Heaviside activation in zeroth-order gradient estimation and how this limits the effectiveness of standard zeroth-order method in SNNs. The paper examines whether exploiting the intrinsic low-dimensional structure of SNN training trajectories can mitigate the variance amplification issue and enable forward-only SNN training on-chip.

**Compliance With Llm Reviewing Policy:**

Affirmed.

**Final Justification:**

Authors addressed my primary concerns in their rebuttal, conducted new experiments and validated their claims. They also provided the missing limitations section. Hence I will update my score to weak accept.

**Key Questions For Authors:**

Could you please clarify whether the subspace is fixed throughout the training or updated online

In section 4.1, I believe you meant to write ANNs instead of SNNs in this line

"Here we examine this property for SNNs to verify that the lowdimension trajectory hypothesis is also true for SNNs."

**Limitations:**

No limitation section provided. Please discuss the weaknesses

**Strengths And Weaknesses:**

Strengths:
- The paper clearly articulates why vanilla ZO methods fail for SNNs, linking the instability to the step-function nature of the Heaviside activation and its variance amplification effect
- Leveraging low-dimensional training trajectories to reduce ZO variance is intuitive and the empirical PCA results show that a small number of principal components explain >90% of the variance. , thus making subspace training feasible.
- Experiments span CIFAR10, CIFAR100, DVS-Gesture and ImageNet1K, and include training from scratch, continual learning and memory comparisons

Weaknesses:
- The subspace is constructed from BPTT training trajectories, which should make one think, does SZO truly eliminate reliance on backpropagation during deployment? Or is the access to a pretrained BP model assumed?
- The lack of literature survey and experiments on existing forward-only learning frameworks such as equilibrium propagation [1].
- While PCA analysis supports the low dimensional hypothesis for classification tasks, it is unclear whether this holds for time series prediction tasks, larger time horizons




[1] EqSpike: Spike-driven equilibrium propagation for neuromorphic implementations

---

> ### Author Rebuttal · Authors · 2026-03-31
>
> We sincerely thank the reviewer for the detailed comments and constructive suggestions. Below we provide clarifications and additional experiments to address the concerns.
> ## 1. The subspace is constructed from BPTT training trajectories.
> Thanks for your comment. Our target scenario is on-chip learning based on a pretrained model. In such settings, subspace can be built offline in the pre-training stage. In the on-device learning stage, learning is performed upon pretrained models solely through forward perturbation-based gradient estimation. A typical application is unsupervised online adaptation in cross-domain scenarios, as presented in the experiments in Sec. 6.3.
>
> This setup echoes the brain’s learning paradigm, where neural architectures and synapse strengths are innately initialized and only modest learning is required to acquire new tasks. We hope our method can inspire efficient, scalable neuromorphic on-chip learning for real-world applications.
> ## 2. The lack of literature survey and experiments on existing forward-only learning frameworks such as equilibrium propagation [1].
> Thanks for your suggestion. Equilibrium propagation based methods, such as EqSpike [1], are based on local energy minimization and provide a BP-free, hardware-friendly learning paradigm. However, these methods have so far been demonstrated on small models and simpler datasets, and their scalability to larger SNNs and more challenging datasets needs further study.
>
> According to this suggestion, we additionally evaluated SZO under the same model and dataset setting as [1] and compare it with EqSpike. As shown in Table 1, both SZO and EqSpike achieve performance close to BP. We will add this comparison in the revised version. Code for reproducing the results in Tables 1 is available at the repository link provided in our paper.
>
> Table 1. Accuracy (%) comparison for training from scratch on MNIST.
> | Dataset| Model | BP | BP-Subspace | SZO-CGE | SZO-RGE(q=20) | EqSpike |
> |-|-|-|-|-|-|-|
> |MNIST | Spike-MLP(784-100-10) |97.57| 97.51 | 97.56 |97.50|96.87|
>
> ## 3. It is unclear whether low dimensional hypothesis holds for time series prediction tasks, larger time horizons.
> Thanks for this constructive comment. According to this comment, we conducted PCA analysis on the training trajectory of SpikF [2] on the Exchange dataset with different prediction lengths, and observed that it also exhibits a low-dimensional structure (see Table 2).  We trained SpikF from scratch using our SZO method within a 20-dimensional subspace, and report the resulting MSE and MAE in Tables 3 and 4, respectively. Both metrics are averaged over prediction lengths of 96, 192, 336, and 720. We will further explore subspace-based ZO training on more time series prediction tasks in future work.
>
> To evaluate the performance under larger time horizons, we conducted additional experiments with substantially larger numbers of timesteps, including DVS-Gesture with 80 timesteps and Spiking Heidelberg Digits (SHD) with 100 timesteps. As shown in Table 5, SZO remains effective under these longer temporal settings.
>
> Code for reproducing the additional results in Tables 2-5 is available at the repository link in our paper.
>
> Table 2. Cumulative explained variance analysis of the training trajectory of SpikF on the Exchange dataset under different prediction length.
> |Prediction length| PC1| PC2| PC3| PC4| PC5| PC6| PC7| PC8| PC9| PC10|
> |-|-|-|-|-|-|-|-|-|-|-|
> | 96 | 0.80| 0.87| 0.90| 0.92 | 0.93 | 0.94 |0.95 | 0.95| 0.96|0.96|
> | 192 |0.75 | 0.84|0.88| 0.91| 0.93| 0.94 |0.96 | 0.96| 0.97|0.98|
> | 336 | 0.81| 0.89 |0.91| 0.92| 0.94| 0.94 |0.95 |0.96|0.96|0.97|
> | 720 | 0.80| 0.88 |0.91| 0.93| 0.95 | 0.96|0.97 |0.97|0.98|0.98|
>
> Table 3. MSE comparison for training from scratch on Exchange dataset.
> | Dataset| Model | BP | BP-Subspace | SZO-CGE | SZO-RGE(q=20) |
> |-|-|-|-|-|-|
> |Exchange| SpikF|0.360| 0.372| 0.363| 0.375  |
>
> Table 4. MAE comparison for training from scratch on Exchange dataset.
> | Dataset| Model | BP | BP-Subspace | SZO-CGE | SZO-RGE(q=20) |
> |-|-|-|-|-|-|
> |Exchange| SpikF|0.402| 0.411 | 0.409 | 0.415 |
>
> Table 5. Accuracy (%) comparison on training from scratch with larger timesteps.
> | Dataset| Timestep ($T$) | BP | BP-Subspace | SZO-CGE | SZO-RGE(q=20) |
> |-|-|-|-|-|-|
> |DVS-Gesture |80 | 96.88 | 96.18 | 95.59 |95.83|
> |SHD |100 | 65.98 | 63.95 | 62.63 | 61.64|
>
> ## 4. Whether the subspace is fixed throughout the training.
> The subspace is fixed throughout training in all tasks. We will clarify this more explicitly in the revised version.
> ## 5. In section 4.1, I believe you meant to write ANNs instead of SNNs.
> Thanks for pointing this out. We will correct this sentence.
> ## 6. No limitation section provided.
> We will add a discussion on the limitations in the revised version, such as the reliance on hand-crafted and pre-constructed subspace.
> ## References
> [1] EqSpike. iScience, 2021
>
> [2] SpikF: Spiking Fourier Network for Efficient Long-term Prediction. ICML, 2025

---

> > ### Author Rebuttal · Reviewer_Kj3r · 2026-04-04
> >
> > I thank the authors for the constructive rebuttal. I appreciate the authors for conducting additional experiments in a short timeframe and I would be willing to raise my score, given some additional concerns are addressed.
> >
> > I understand the intended setting of starting with pretrained models but the abstract might be misleading. Specifically it's this statement "whereas BP is computationally and memory intensive that unsuitable for on-chip edge learning" that creates confusion since it might be interpreted that there is no BP involved at all. In my opinion, the scope should be made clear in the abstract.
> >
> > While it is true that methods like EqProp haven't been scaled to challenging tasks yet, the approach does not rely on pretraining models. Nevertheless, I thank the authors for conducting the experiments on EqProp and with the DVS Gesture dataset.
> >
> > Could you please provide the limitation section that you intend to add in the main text, so that the reviewers could review it before you include it?

---

> > > ### Author Response · Authors · 2026-04-04
> > >
> > > We sincerely thank the reviewer for the further comments. Below we provide responses to address your concerns.
> > > ## (1) I understand the intended setting of starting with pretrained models but the abstract might be misleading. Specifically it's this statement "whereas BP is computationally and memory intensive that unsuitable for on-chip edge learning" that creates confusion since it might be interpreted that there is no BP involved at all. In my opinion, the scope should be made clear in the abstract.
> > > Thank you for pointing this out. We agree that the original statement in the abstract could be misleading, as it may be interpreted as suggesting that BP is not involved at any stage. We will revise the abstract to more precisely reflect the scope of our work as follows:
> > > *The human brain is a biologically instantiated on-device neural system that integrates both learning and inference in a unified architecture, which enables rapid and flexible learning on-the-fly. **This extraordinary online learning ability is realized through biological learning mechanisms operating on a well-initialized innate model. This work considers the on-chip edge learning upon pretrained models with zeroth-order (ZO) methods. ZO optimization methods, which resemble bio-plausible perturbation-based learning, offer a promising approach that enables learning with only forward passes and hence can significantly reduce the complexity of on-chip hardware implementation.** However, in this work we show that applying ZO methods to spiking neural networks (SNNs) is non-trivial due to the step-function nature of spiking activation...*
> > > ## (2) While it is true that methods like EqProp haven't been scaled to challenging tasks yet, the approach does not rely on pretraining models.
> > > Thank you for this clarification. We agree that EqProp differs fundamentally from our setting, since it does not rely on pretrained models and is BP-free that only relies on local computation. Such bio-plausible local learning methods are promising for on-chip learning due to their hardware efficiency and low energy consumption. We will add a discussion in the related work to acknowledge these advantages.
> > > ## (3) Provide the limitation section that you intend to add in the main text.
> > > Thanks for the suggestion. We will add a section to discuss the limitations of our work as follows:
> > > *In the proposed SZO method, the subspace is constructed in a hand-crafted manner from offline BP pretraining. While this design is effective in our experiments, its effectiveness depends on the transferability of the resulting subspace to downstream on-chip learning tasks. Exploring data-driven or meta-learning based approaches can be expected to construct more transferable subspace. Moreover, our study mainly focuses on small-to-medium-scale models and classification tasks. Extending the proposed method to larger models and tasks such as sequential language modeling is an important direction for future work.*
> > >
> > > Finally, we sincerely appreciate the reviewer’s valuable comments, which are constructive and helpful. In particular, we are grateful for the suggestions to clarify the role of the BP pretraining stage in obtaining the subspace, to compare with existing forward-only learning methods such as EqSpike, to evaluate on time series prediction tasks and tasks with larger time horizons, and to revise the Abstract to better clarify the scope of the paper. These suggestions help us improve the paper substantially. We thank the reviewer for the careful review of our paper.

---

### Decision · Program_Chairs · 2026-04-30

**Decision:**

Accept (regular)

**Comment:**

This paper proposes SZO, a subspace-based Zeroth-Order (ZO) optimization method designed for Spiking Neural Networks (SNNs). The work investigates the variance amplification effect caused by the Heaviside activation function and demonstrates how leveraging the low-dimensional structure of training trajectories can enable efficient forward-only on-chip learning.

Following the rebuttal, the paper received final scores of 4, 5, 2, and 4. While there was a rejection score, it is notable that the reviewer who gave the low score (Reviewer abx4) explicitly acknowledged that the authors had resolved all their technical questions during the discussion. However, they ultimately maintained their low score based on their stance regarding the reliance on offline BP for subspace construction.

Despite this disagreement, the majority of the reviewers recognize the technical solidity of the work. The proposed method offers a significant step toward neuromorphic on-chip learning by enabling fast adaptation without backpropagation. Therefore, I recommend acceptance.